# PSMA-Targeted Radiolabeled Peptide for Imaging and Therapy in Prostate Cancer: Preclinical Evaluation of Biodistribution and Therapeutic Efficacy

**DOI:** 10.3390/ijms26157580

**Published:** 2025-08-05

**Authors:** Ming-Wei Chen, Yuan-Ruei Huang, Wei-Lin Lo, Shih-Ying Lee, Sheng-Nan Lo, Shih-Ming Wang, Kang-Wei Chang

**Affiliations:** 1Department of Isotope Application Research, National Atomic Research Institute, Taoyuan 32546, Taiwan; brian0512@nari.org.tw (M.-W.C.); yuanruei@nari.org.tw (Y.-R.H.); loweilin@nari.org.tw (W.-L.L.); shihying@nari.org.tw (S.-Y.L.); loshengnan@gmail.com (S.-N.L.); funnyjoe@nari.org.tw (S.-M.W.); 2Taipei Neuroscience Institute & Laboratory Animal Center, Taipei Medical University, Taipei 110301, Taiwan

**Keywords:** theranostic agent, lutetium-177

## Abstract

Albumin-binding agents enhance tumor uptake of radiopharmaceuticals targeting prostate-specific membrane antigens (PSMAs) in radiotherapy. We synthesized PSMA-NARI-56, a molecule with both PSMA targeting activity and albumin-binding moiety, labeled with ^177^Lu as the therapeutic agent. The aim of this study was to determine the specific binding of ^177^Lu-PSMA-NARI-56 towards PSMA, assess its biodistribution, and evaluate therapeutic effectiveness by tumor-bearing mice. The effect of ^177^Lu-PSMA-NARI-56 viability of PSMA-positive cell (LNCaP) was evaluated. Biodistribution and endoradiotherapy studies were utilized to determine the distribution, targeting, and anti-tumor efficacy by tumor-bearing mice identified by ^111^In-PSMA-NARI-56. ^177^Lu-PSMA-NARI-56 exhibited a significant impact on the viability of the LNCaP cell. Biodistribution results revealed the maximum tumor uptake of ^177^Lu-PSMA-NARI-56 occurring within 24 h, reaching 40.56 ± 10.01%ID/g. In radionuclide therapy, at 58 days post-injection (p.i.), ^177^Lu-PSMA-NARI-56 demonstrated superior tumor inhibition (98%) compared to ^177^Lu-PSMA-617 (58%), and the mouse survival rate after 90 days of radiotherapy (90%) was also higher than that of ^177^Lu-PSMA-617 (30%) in LNCaP tumor-bearing mice. In the PSMA-positive animal model, ^177^Lu-PSMA-NARI-56 shows higher potential radiotheranostic and prolonged accumulation (identify by ^111^In-PSMA-NARI-56/nanoSPECT/CT image), offering the potential for improved treatment effectiveness and increased survival rates when compared to ^177^Lu-PSMA-617.

## 1. Introduction

Prostate cancer, particularly common in developed countries (a cancer unique to men, the incidence and mortality rates are in an increasing upward trend), holds a prominent position as the second most frequently diagnosed cancer among men and the fourth most common cancer worldwide, thus making it the leading malignant tumor in males [1]. There are various therapeutic approaches available for the management of prostate cancer-like androgen deprivation therapy (ADT), but over time, the disease may develop resistance, resulting in a condition referred to as castration resistance [1]. This state poses a considerable obstacle for patients with metastatic castration-resistant prostate cancer (mCRPC) [2]. Since 2004, a variety of medicines have been approved for the treatment of metastatic castration-resistant prostate cancer (mCRPC). The use of these medications depends on factors such as the extent of metastatic tumor burden, prior treatment history, and the expected safety profile. These treatments have contributed to improvements in both overall survival and progression-free survival (PFS) [2,3,4,5,6]. In order to address these challenges, the integration of precise cancer imaging and therapeutic interventions holds significant promise in the realm of personalized medicine for imperative mitigation of patients’ adverse effects, and the optimization of therapeutic efficacy [7].

Prostate-specific membrane antigen (PSMA) is a type II transmembrane glycoprotein (also referred as folate hydrolase I or glutamate carboxypeptidase II) consisting of 750 amino acids, and possesses both intracellular and extracellular domains [8,9,10,11]. PSMA is known to be present on both benign prostate epithelium as well as prostate cancer cells and has been observed in various other tissues, including the kidneys, small intestine, and salivary glands [8,11]. The expression of prostate cancer cells is significantly greater than that of normal tissues. Notably, the heightened expression of PSMA is correlated with the presence of more aggressive prostate tumors, including hormone-refractory cancers [12]. Hence, PSMA holds significant potential as a viable target for both diagnostic imaging and therapeutic interventions in individuals afflicted with prostate cancer.

For radiotherapy, several isotopes have been developed in combination with PSMA-targeting agents. Commonly used isotopes include Iodine-131 (^131^I, a β-emitter with a half-life of 8.03 days), Yttrium-90 (^90^Y, a β-emitter with a half-life of 2.66 days), Lutetium-177 (^177^Lu, a β-emitter with a half-life of 6.65 days), and Actinium-225 (^225^Ac, an α-emitter with a half-life of 9.95 days). These different radioisotopes can be utilized in therapeutic applications of PSMA-targeting agents, allowing for tailored radiotherapy strategies based on emission type, energy, and tissue penetration properties to optimize treatment efficacy and minimize off-target effects. These isotopes have found extensive application in the radiotherapy field. Ga-68-PSMA-11 has been approved in the United States by the Food and Drug Administration (FDA) as the first ^68^Ga-radiopharmaceutical for the PET imaging of PSMA-positive prostate cancer in 2020. In March 2022, FDA approved Lutetium-177 PSMA-617 (as Lutetium-177-PSMA or ^177^Lu-vipivotide tetraxetan, Pluvicto) as a radioligand target with PSMA, to become the treatment for metastatic, castration-resistant, and refractory prostate cancer in recent years. In August 2022, this medication received marketing authorization in the United Kingdom [13,14,15]. In Fluorine-18-related compounds, two newer fluorine-18 agents’ (pylarify (piflufolastat) approval on 27 May 2021 and POSLUMA’s (flotufolastat) approval on 25 May 2023) also serve to quantify PSMA expression for the enrollment of patients in PSMA radiotherapy and to stage disease progression. Notwithstanding these achievements, a considerable number of patients exhibit an inadequate response to radioligand therapy and encounter disease progression either during or after treatment. To mitigate off-target radiation exposure to non-malignant organs, such as the kidneys and salivary glands, strategies involving PSMA inhibitors have been explored. One notable advancement is the use of JHU-2545, a prodrug of the PSMA inhibitor 2-PMPA, which selectively delivers the active compound to the kidneys and salivary glands. Preclinical and early clinical studies demonstrated that JHU-2545 can significantly reduce the uptake of radiolabeled PSMA ligands in these organs—by up to 85%—while preserving tumor targeting, thereby improving the therapeutic window of PSMA-targeted radioligand therapy. This selective shielding approach represents a promising direction for reducing treatment-related toxicities without compromising efficacy [15].

In recent times, utilizing either unaltered or modified albumin-binding motifs to enhance the circulation duration of pharmaceuticals and optimize their uptake by tumors for maximizing compounds’ therapeutic potential binders has emerged as a compelling approach in the development of radionuclide therapy agents [15,16,17,18,19]. Some of the drawbacks experienced with Pluvicto chiefly concern high kidney and salivary gland uptake that can lead to loss of function in those organs. In our previous study, we effectively synthesized a PSMA-targeting compound known as PSMA-NARI-56. This molecule was designed for possessing long-circulating properties, achieved through the integration of albumin-binding agents, and also a conjugated DOTA chelator was incorporated into the compound to facilitate radiolabeling with radioactive isotopes. This helped to ensure more tumor uptake and less uptake in other organs to maximize therapeutic effect and decrease side effects/toxicities. In previous studies, we have successfully utilized the integration of albumin-binding motifs into PSMA-targeted compounds to significantly enhance blood retention and tumor accumulation [20]. In this study, we performed an early effectiveness evaluation to determine the effects of the medication on the murine tumor model and we also delve deeper into novel data, encompassing cell viability assays, endoradiotherapy investigation, and biodistribution analysis. The inclusion of these data sources is anticipated to yield more profound insights into our research and perhaps furnish substantial validation for drug development and clinical applications.

## 2. Results

### 2.1. LNCaP Cell Viability Assay with ^177^Lu-PSMA-NARI-56

Radio-HPLC and radio-thin layer chromatography (TLC) analysis (radiopurity > 95%) confirmed the efficient labeling of ^177^Lu-PSMA-NARI-56, and subsequent experiments were carried out to assess LNCaP cell viability. As shown in Figure 1, the effects of different concentrations of ^177^Lu-PSMA-NARI-56 (0.1, 0.3, and 1 MBq/mL) on the viability of LNCaP cells were evaluated. The results showed that at concentrations of 3 and 10 MBq/mL, the effects on cell viability were significantly different (*p* < 0.01 at 3 MBq/mL and *p* < 0.001 at 10 MBq/mL). A comparison of the effects of ^177^Lu-PSMA-NARI-56 on the viability of LNCaP and PC-3 cells showed a significant difference (*p* < 0.001) at all administrated concentrations.

### 2.2. Mice NanoSPECT/CT Images

NanoSPECT/CT images of ^111^In-PSMA-NARI-56 in LNCaP and PC-3 tumor-bearing mice are shown in Figure 2A,C, respectively. Figure 2A illustrates the accumulation state of ^111^In-PSMA-NARI-56 in the LNCaP mice and shows excellent activity in the tumor and appropriate biological metabolisms 24 h post-injection and a significant signal in the tumor up to 96 h. However, the tumors in the competitively binding radionuclide group (Figure 2B) and PC-3 tumor-bearing mice showed only a slight uptake, and there was no significant signal accumulation in the tumors after 24 h (Figure 2C). Relatively weak uptakes by the tumor were observed after approximately 4 h (Figure 2A–C). Regarding the competitive assay, a semiquantitative assessment of the images acquired using nanoSPECT/CT, from mice with LNCaP tumors, was performed (Figure 2B). The tumor uptake values at 1, 4, 24, 48, 72, and 96 h post-injection were determined to be 4.73 ± 0.69, 4.84 ± 0.19, 2.96 ± 1.54, 2.16 ± 0.66, 1.69 ± 0.26, and 1.27 ± 0.16% ID/g, respectively.

### 2.3. Biodistribution of ^177^Lu-PSMA-NARI-56 in LNCaP Xenografts Tumor-Bearing Mice

Table 1 presents the biodistribution analysis of ^177^Lu-PSMA-NARI-56 in mice with LNCaP tumor xenografts. The biodistribution of ^177^Lu-PSMA-NARI-56 was characterized at different time points after its administration. Blood clearance decreased to a concentration of 0.13 ± 0.02% ID/g at 96 h after administrations. The initial kidney uptake was 107.65 ± 37.19% ID/g at the 1 h time point, decreasing to 1.55 ± 0.26% ID/g by 96 h. The observed tumor accumulation exhibited a compelling pattern. The initial uptake of the substance was 26.52 ± 12.11% ID/g at the 1 h time point. Subsequently, a gradual increase in uptake was observed, culminating in 40.56 ± 10.01% ID/g at 24 h after injection. Subsequently, the concentration progressively declined to 13.41 ± 2.89% ID/g after 96 h. In contrast, the liver exhibited relatively low levels of uptake, as evidenced by 7.15 ± 1.73% ID/g at the 1 h time interval, which significantly decreased to 0.10 ± 0.02% ID/g at the 96 h mark.

### 2.4. Efficacy of ^177^Lu-PSMA-NARI-56 vs. ^177^Lu-PSMA-617 Radionuclide Therapy

The effectiveness of ^177^Lu-PSMA-NARI-56 radionuclide therapy was assessed in LNCaP tumor-bearing mice. Alterations in the tumor dimensions upon intravenous administration of the designated drugs through tail vein injection are shown in Table 2. The measurements were performed over a period of 58 days. Overall, the tumor size exhibited progressive growth in the mice belonging to the control group from days 0 to 60. By the 58th day, the %TGI for ^177^Lu-PSMA-NARI-56 was recorded as 98%, whereas for ^177^Lu-PSMA-617, it stood at 58%. The mice administered with ^177^Lu-PSMA-617 exhibited an initial inhibition of tumor growth, followed by a subsequent upward trajectory beyond the 30th day in the non-treatment group. In contrast, tumor growth in the cohort of mice that received ^177^Lu-PSMA-NARI-56 was substantially inhibited throughout the duration of the study.

This study also used ^111^In-PSMA-NARI-56/nanoSPECT/CT imaging, as shown in Figure 3, to evaluate the tumor dimensions in mice treated with ^177^Lu-PSMA-NARI-56 at specific time points (−day 2 (two days prior to experiment), day 25, day 39, and day 60). Significantly, the findings through direct measurements and nanoSPECT/CT images revealed that a considerable proportion of the mice had complete eradication of the tumor.

As shown in Figure 3, mortality trends were monitored up to 90 days, and the survival curves are visually depicted in Figure 4. Based on predetermined euthanasia criteria, tumors in the control group were found to rapidly attain the required size for euthanization, leading to the sacrifice of all subjects by day 76 of the study. Upon receiving treatment with ^177^Lu-PSMA-617, the mice harboring tumors exhibited signs that met the euthanization criteria starting from the 48th day after study commencement. At 90 days following the commencement of the experiment, a mere 30% of the mice under investigation had managed to survive. In stark contrast, the experimental group of mice that received treatment with ^177^Lu-PSMA-NARI-56 demonstrated a notable survival rate of 90% at the 90-day mark.

The survival curves for the three groups were formally compared using a Log-Rank test, which revealed a statistically significant overall difference between the groups (χ^2^ = 14.179, *p* < 0.001). Subsequent pairwise comparisons using the Holm–Sidak method confirmed that treatment with ^177^Lu-PSMA-NARI-56 resulted in significantly longer survival compared to the control group (*p* < 0.001) and the ^177^Lu-PSMA-617 group (*p* = 0.008). While the ^177^Lu-PSMA-617 group demonstrated a modest prolongation in survival relative to controls, the different was not statistically significant (*p* = 0.335) in Figure 4.

## 3. Discussion

The overexpression of PSMA in cells affected by prostate cancer presents a promising and innovative focus for the development of theranostics in the field of prostate cancer. ^177^Lu-PSMA-617 therapy represents a feasible therapeutic approach for mCRPC [20,21]. The rapid clearance of this peptidic molecule from the circulation necessitates the administration of high doses and frequent clinical treatments in radiotherapy. Consequently, this leads to elevated clinical expenses and systemic toxicity. The use of small-molecule albumin binders to extend the circulation time of pharmaceuticals and maximize their tumor uptake has become an attractive strategy for the design of endoradiotherapeutic agents.

In a previous study, we synthesized a compound by conjugating an albumin-binding molecule and a DOTA chelator onto PSMA-617 Appendix A. This resulting compound, known as PSMA-NARI-56, was subsequently labeled with ^177^Lu and subjected to preliminary analysis using animal imaging techniques Appendix A [22]. However, to comprehensively evaluate the clinical potential of this compound, it remains crucial to assess its impact on the viability of specific cancer cells, especially given the association of PSMA with the majority of prostate cancer cells. Our studies utilized PC-3, a cell line that has been genetically modified to exhibit a significantly reduced level of PSMA expression in comparison to LNCaP cells [23,24,25,26]; our choice of the LNCaP cell line is rooted in its closer resemblance to natural disease onset and its pathological characteristics akin to real-case scenarios, making it more representative. By conducting cell viability assays in PC-3 and LNCaP cells (Figure 1), the results show non-specific binding with ^177^Lu on LNCaP cell and ^177^Lu-PSMA-NARI-56 on PC-3 cell, and we deepen our understanding of how this drug has specific effects on special cells and enhances its therapeutic role due to the albumin in extended prostate cancer treatment.

To minimize potential harm to normal tissues, it is imperative to assess the in vivo specificity of the medicine, ensuring that it exclusively targets the intended tissue or cell. Previous studies have compared the PSMA-NARI-56 marker labeled with ^111^In and ^177^Lu, confirming similar targeting profiles. In this study, two in vivo assays were conducted to validate the specificity of PSMA-NARI-56 for the PSMA receptor. These included a competitive binding experiment, where mice bearing LNCaP tumors received either ^111^In-PSMA-NARI-56 alone (Figure 2A) or co-administered with a 100-fold molar excess of unlabeled PSMA-NARI-56 (Figure 2B), and a second assay using PC-3 tumor-bearing mice, which lack PSMA expression (Figure 2C). The competitive assay showed a peak tumor uptake of ^111^In-PSMA-NARI-56 at 4 h post-injection, with approximately 4.84 ± 0.19% ID/g. However, in the competition group, both tumor and kidney signals were significantly reduced, suggesting PSMA-specific binding, and that kidney uptake is at least partially PSMA mediated. Co-administration with albumin appeared to prolong systemic distribution and reduce renal accumulation. In PC-3 tumors, only a modest uptake peak was observed at 4 h, followed by rapid clearance, further supporting the PSMA-specific accumulation of PSMA-NARI-56.

Based on the NanoSPECT/CT imaging and semiquantitative data, ^111^In-PSMA-NARI-56 demonstrates excellent tumor specificity and prolonged retention in PSMA-positive LNCaP tumors, with significant radioactivity accumulation sustained up to 96 h post-injection and high initial uptake values of around 4.7–4.8% ID/g within the first 4 h, indicating strong target affinity and favorable imaging characteristics. However, a noticeable decline in tumor retention was observed after 24 h, decreasing from 2.96% ID/g to 1.27% ID/g by 96 h. Although ^111^In-PSMA-NARI-56 exhibits high PSMA-specific binding—as confirmed by reduced uptake in both competitive binding and PSMA-negative PC-3 tumors—these findings suggest that enhancing intratumoral retention, perhaps through modifications to improve internalization or intracellular trapping, could further optimize its therapeutic efficacy.

In Table 1, the distribution of ^177^Lu-PSMA-NARI-56 in LNCaP tumor-bearing mice was compared with ^177^Lu-PSMA-56. ^177^Lu-PSMA-NARI-56 was also shown to have a high uptake rate in tumors. The persistently high tumor uptake of PSMA-NARI-56 is attributed to slower blood clearance and enhanced interactions with albumin, which reduce the steric accessibility of PSMA ligands by metabolizing enzymes, and thus may have higher metabolic stability. In comparison to previous studies, radiolabeled PSMA-NARI-56 had higher blood activity after 1 h compared to PSMA-617, which may have contributed to its delivery to tumors over time. In terms of tumor uptake, the tumor uptake of PSMA-NARI-56 was approximately 1.8-fold higher after 1 h than that of PSMA-617. The highest tumor uptake of PSMA-NARI-56 (40.56 ± 10.01%ID/g) was observed after 48 h, and the tumor uptake of PSMA-NARI-56 was also 2.4-fold higher than that of PSMA-617 after 72 h, which confirms the effect of higher plasma protein (albumin) binding on PSMA-expressing tissue uptake. Similar results have been observed by other research groups, with albumin-bound ligands exhibiting higher tumor accumulation and stronger retention [20,27,28]. Prolonged ^177^Lu- PSMA-NARI-56 intra-irradiation therapy applications may deliver higher radiation doses on target tissues and similar results of high PSMA-performing kidney accumulation, likely limiting the maximum dose during treatment, and possible potential nephrotoxicity or hematologic toxicity is something to consider [29]. However, unlike the human kidney, which only moderately expresses PSMA, the mouse kidney is highly expressive of PSMA [30], and therefore absorbed doses in the human kidney derived from mouse renal uptake data may be overestimated. In addition, the bio-distribution experiments show that post-administrated of ^177^Lu-PSMA-NARI-56, the accumulated radiation dose amount in target organs should not cause cumulative damage in the body. Secondly, the administration of lysine and arginine has been shown to reduce nephrotoxicity in clinical practice [31]. Current preclinical studies have also begun to use 2-(phosphonomethyl) pentanedioic acid (PMPA) and mannitol to reduce renal uptake of PSMA-targeted radiopharmaceuticals [32,33], and these will be the strategies that will be attempted in future studies.

In our previous study, we attempted to administer ^177^Lu-PSMA-NARI-56 and ^177^Lu-PSMA-617 to LNCaP tumor-bearing mice to assess the treatment efficiency by observing the changes in tumor size in the mice. On day 44 after treatment, the TGI of ^177^Lu-PSMA-NARI-56 or ^177^Lu-PSMA-617 was 96.0% and 54.7%, respectively [26]. Continuing from the previous trial and extending the treatment observation period to 58 days, the TGI reached 98% for ^177^Lu-PSMA-NARI-56 and 58% for ^177^Lu-PSMA-617, both of which showed an increase in tumor inhibition, although there was a greater difference in the size of the tumors in the mice treated with ^177^Lu-PSMA-617. In this study, the tumor size in the control group continued to increase after treatment (saline injection), and the median survival of the control group was 58 days (the mice were euthanized when they died of natural causes or when the tumor size reached 2000 mm^3^ or when the tumor weight was more than 1/10th of the body weight of the mouse); in the group treated with ^177^Lu-PSMA-617, the median survival was 62 days as Figure 3A,B. Our study shows some differences from the results reported by Kuo et al., which may be attributed to variations in tumor size [32]. In our experiment, the control group mice had an initial tumor size of 191 ± 15 mm^3^, while the tumor sizes in the ^177^Lu-PSMA-NARI-56 and ^177^Lu-PSMA-617 groups were 312 ± 91 mm^3^ and 326 ± 98 mm^3^, respectively. In Kuo et al.’s experiment, mouse tumor sizes ranged from 531 to 640 mm3 [18,23]. Consequently, even with the administration of the same 18.5 MBq of radiopharmaceutical therapy, the larger tumors and more stringent conditions (euthanasia of mice when tumor volume reached 1000 mm^3^) resulted in differences in the median survival between the control group and the ^177^Lu-PSMA-617 group. According to these results, ^177^Lu-PSMA-NARI-56 shows more excellent tumor growth inhibition than ^177^Lu-PSMA-617. A clinical dose of approximately 18.5 MBq of Lu-177-PSMA has been established to optimize the balance between efficacy and safety in treating metastatic castration-resistant prostate cancer (mCRPC). This dosing ensures consistent radiation delivery across patients, maximizing tumor targeting while minimizing off-target effects to organs such as the kidneys and salivary glands. Clinical trials had demonstrated that this regimen provides effective tumor control with acceptable toxicity.

In radionuclide therapy study (Figure 4), the ^177^Lu-PSMA-NARI-56 group exhibited promising therapeutic efficacy. At day 90, only 10% of the mice were sacrificed due to tumors exceeding 2000 mm^3^, resulting in an overall survival rate of 90%. However, 70% of the mice in this group showed complete tumor regression. In comparison, the ^177^Lu-PSMA-617 group had a survival rate of only 30% at day 90, although the tumors in these 30% of mice also completely regressed. Throughout the experiment, no mice were excluded from the survival curve due to significant changes in body weight or natural death, suggesting that the impact of the 18.5 MBq activity level of ^177^Lu-PSMA-NARI-56 on mouse kidneys was limited. It has been reported that 120 MBq of ^177^Lu-PSMA-617 at an activity level is most effective in suppressing tumor growth [32]. This study, however, indicates that in the murine model of prostate cancer tumors, at relatively lower activity levels, ^177^Lu-PSMA-NARI-56 demonstrates superior therapeutic efficacy and higher survival rates compared to ^177^Lu-PSMA-617.

## 4. Materials and Methods

### 4.1. Radio-Synthesis of Peptides with Indium-111 and Lutetium-177

Chloride salts of Indium-111 (^111^In) and Lutetium-177 (^177^Lu) were procured from National Atomic Research Institute (NARI) (Taoyuan, Taiwan) and ITM Medical Isotopes GmbH (Garching/Munich, Germany), respectively. Peptides PSMA-NARI-56 and PSMA-617 were sourced from Ontores Biotechnologies Co., Ltd. (Hangzhou, China).

Regarding the preparation of ^111^In-PSMA-NARI-56, 13.9 nmol of the peptide (dissolved in DMSO) was heated with 0.23–0.26 GBq of ^111^InCl_3_ in a solution of 1 M sodium acetate (pH 6.0) at 95 °C for 15 min, a similar procedure for ^177^Lu-PSMA-NARI-56, except the peptide was incubated with 0.47–0.51 GBq of ^177^LuCl_3_ in a 0.4 M sodium acetate solution (pH 5.0) for 30 min at 95 °C. Labeling efficiency was subsequently analyzed through an AR-2000 radio-TLC imaging scanner (Bioscan, France) using 10% methanol and radio high-performance liquid chromatography (radio-HPLC) equipped with a UV detector (280 nm) (Water T3 C18 column (3.5 μm, 80 A, 4.6 × 250 mm) (mobile buffer is acetonitrile + 0.1% trifluoroacetic acid (TFA) 1mL/min) in ^111^In-labeling and with 0.1 M citric acid in ^177^Lu-labeling, as seen in previous studies [19].

In ^177^Lu-PSMA-617 radiolabeling was mixed 19.2 nmol of peptides (dissolved in water) with 0.47–0.51 GBq of ^177^LuCl_3_, which is the same procedure as ^177^Lu-PSMA-NARI-56. The specific activity of ^111^In-PSMA-NARI-56, ^177^Lu-PSMA-NARI-56, and ^177^Lu-PSMA-617 was found to be 16.5–18.7, 33.8–36.7, and 24.5–26.6 GBq/μmole, respectively.

### 4.2. Cell Culture and Tumor-Bearing Mice

The PSMA regenerative LNCaP cell (positive) and PC-3 cell (negative) were procured from the Bioresource Collection and Research Center. The LNCaP and PC-3 cells were cultured in RPMI 1640 and F-12K medium, supplemented with 10% (*v*/*v*) fetal bovine serum, along with 100 units/mL penicillin and 100 μg/mL streptomycin. The culture conditions included a humidified atmosphere at 37 °C with a CO_2_ concentration of 5%.

Six-week-old male BALB/c nude mice were sourced from BioLASCO Taipei, Taiwan Co., Ltd. The mice were kept in a regulated environment with a 12 h light cycle at 22 °C and free access to food and water. All animal experimental procedures were sanctioned by the NARI Institutional Animal Care and Use Committee (IACUC-2020-0077). For generating LNCaP and PC-3 tumor xenografts, we, respectively, implanted 1 × 10^7^ LNCaP cells and 2 × 10^6^ PC-3 cells into the forelegs of the BALB/c nude mice. For the evaluation of therapeutic efficacy, tumor volumes ranged approximately between 200 and 600 mm^3^ and 150 and 450 mm^3^, respectively.

### 4.3. Cell Viability Assay

The effect of ^177^Lu and ^177^Lu-PSMA-NARI-56 on the viability of LNCaP and PC-3 tumor cells was determined using the alamarBlue assay, the methodology for which was adapted from the research published by Müller [24]. Both LNCaP and PC-3 cells were populated into 96-well plates at a density of 1 × 10^4^ cells per well and nurtured for 24 h until they reached 60–70% confluence. Subsequent to a single wash with PBS, cells were incubated in 200 μL of serum-free medium infused with either ^177^Lu or ^177^Lu-PSMA-NARI-56, adjusting the radioactivity to 0.1, 0.3, 1, 3, and 10 MBq/mL. After a 4 h incubation period, cells were rinsed once with PBS and then grown in a medium with serum. At the 24 and 48 h mark post-incubation, cells were washed with PBS and subjected to alamarBlue Cell Viability Reagent (Invitrogen, Waltham, MA, USA), 10 μL reagent in 100 μL medium) for another 4 h incubation period. Cell survival was determined by measuring either the fluorescence (Ex/Em: 560 nm/590 nm) or the absorbance (570 nm) using an enzyme-linked immunosorbent plate reader (anthos Zenyth 3100).

### 4.4. NanoSPECT/CT Imaging of ^111^In-PSMA-NARI-56 on Tumor-Bearing Mice

The NanoSPECT/CT Plus scanner system (Mediso Medical Imaging Systems; Budapest, Hungary) was used, during which mice were subjected to 1–2% isoflurane anesthesia to maintain stability. High-resolution NanoSPECT images were captured using nine multi-pinhole gamma detectors equipped with high-resolution collimators. Parameters such as an energy window set to 171 and 245 keV ± 10%, an image dimension of 256 × 256 pixels, and a field of view of 60 × 100 mm were set for optimal imaging.

In ^111^In-PSMA-NARI-56 nanoSPECT/CT imaging, LNCaP and PC-3 tumor-bearing mice and NARI in competitive binding study, mice were pre-treated with a 100-fold molar excess (250 μg) of PSMA-NARI-56 via injection into the tail vein 15 min before receiving the primary radiolabeled peptide (dose between 19.4 and 20.7 MBq, NanoSPECT/CT image acquisition 30 min post-distribution 30 min).

To ascertain the organ-specific uptake values for ^111^In-PSMA-NARI-56, SPECT data were reconstructed via HiSPECT NG software (Göttingen, Germany) and subsequently merged with CT datasets employing InVivoScope software (Bioscan Inc., Washington, DC, USA). All the processed data were examined using PMOD Version 3.3 (PMOD Technologies Ltd., Zurich, Switzerland). The volume of interest (VOI), consisting of the tumor and reference source, was defined using corresponding CT images. These VOI outlines were then transposed onto SPECT images to extract count values for both the tumor and the reference source. The activity range set for the VOI was corresponding to the injected dose per organ gram. Reference source radioactivities were established as 0.8 MBq for ^111^In.

### 4.5. Biodistribution of ^177^Lu-PSMA-NARI-56 in LNCaP Tumor-Bearing Mice

The biodistribution of ^177^Lu-PSMA-NARI-56 was based on the procedures outlined in earlier studies by Lo et al. for the biodistribution component [25]. For the biodistribution examination, a cohort of 24 LNCaP tumor-bearing mice was divided into six subsets (post-injected for 1, 4, 24, 48, 72, and 96 h), each containing four mice. They were all subjected to an intravenous injection that comprised roughly 1.85 MBq and 0.25 μg of ^177^Lu-PSMA-NARI-56. These mice were then put down compassionately via CO_2_ asphyxiation at each designated time, relevant organs were obtained. The extracted organ samples were then carefully rinsed with saline, blotted to dryness, weighed, and their radioactivity was determined with a PerkinElmer 2480 gamma counter (Revvity, Inc., Waltham, MA, USA) The compiled data were expressed as the mean ± standard deviation (SD) of the percentage of the initial dose present in a gram of tissue (% ID/g).

### 4.6. Radionuclide Therapy Study of ^177^Lu-PSMA-NARI-56 and ^177^Lu-PSMA-617

The LNCaP mice were randomly assigned to three treatment groups: ^177^Lu-PSMA-NARI-56 (*n* = 10), ^177^Lu-PSMA-617 (*n* = 10), and a non-treatment group (*n* = 5) through one single intravenous injection of either ^177^Lu-PSMA-NARI-56, ^177^Lu-PSMA-617, and normal saline. The dosage for each injection consisted of 18.5 MBq of ^177^Lu and 7.3 ng/mouse of either PSMA-NARI-56 or PSMA-617. The tumor volume was evaluated on a weekly basis utilizing a digital caliper while concurrently documenting the survival rate of the rodent population. The tumor’s volume was determined through the application of the formula V = (Y × W^2^)/2 to its measured dimensions. In this context, Y and W were denoted as the larger and smaller perpendicular diameters, respectively. The data presented in this study are reported as the mean value, accompanied by the standard error of the mean (SEM). The calculation of the tumor growth inhibition percentage (% TGI) is performed in the following manner: The formula used in this study to calculate the relative change in tumor volume is as follows: [1 − ((relative tumor volume (RTV) of treatment)**/**(RTV of control))] × 100. RTV represents the initiation of treatment and after a duration of 58 days. This value is then multiplied by 100 to express the change as a percentage [19].

Concomitant with the assessment of therapeutic effectiveness, the survival rate of the corresponding murine subjects was monitored. Following the conclusion of the assessment of anti-tumor efficacy on day 58, the mice were subjected to continuous monitoring of their body weight and tumor dimensions until day 90, with any instances of mortality being duly documented. The survival curve was determined based on the following criteria: (1) mortality, (2) euthanizing the mouse if the tumor weight exceeded 1/10 of its body weight, and (3) sacrificing the mouse if the tumor size exceeded 2000 mm^3^.

### 4.7. Statistics

Data points represent mean values and error bars represent standard deviations. Sta-tistical analyses were performed using the unpaired two-tailed Student’s *t*-test, unless otherwise noted. Survival curves were compared using the Log-Rank (Mantel–Cox) test with the Holm–Sidak method for multiple comparisons. *p* values less than 0.05 were considered statistically significant. Significance levels in graphs are marked as follows: ** for *p* < 0.01, and ***, +++ for *p* < 0.001.

## 5. Conclusions

The most significant difference between ^177^Lu-PSMA-NARI-56 and ^177^Lu-PSMA-617 was observed on day 90, where the ^177^Lu-PSMA-NARI-56 group showed a 90% survival rate and 70% complete tumor regression, compared to only 30% survival in the ^177^Lu-PSMA-617 group. This highlights the superior efficacy of ^177^Lu-PSMA-NARI-56 under the same treatment conditions, likely due to its prolonged circulation and enhanced tumor retention.

Regarding the present market availability of Pluvicto, a crucial task ahead involves evaluating the capacity of existing worldwide nuclear medicine facilities to effectively address the anticipated demand resulting from the new concepts of PSMA-617 treatment and theranostics. Therefore, improving the long-term efficacy of the drug and reducing the number of treatments for patients is a feasible direction. Our study provided evidence that ^177^Lu-PSMA-NARI-56 exhibited superiority over ^177^Lu-PSMA-617 when administered with the same radioactivity dosage. This superiority was observed in terms of its enhanced capacity to prevent tumor growth as well as its ability to extend the survival time with ^177^Lu-PSMA-NARI-56. The preclinical data presented a compelling basis for the potential applicability of ^177^Lu-PSMA-NARI-56 in the clinical treatment of prostate cancer in the years to come.

## Figures and Tables

**Figure 1 ijms-26-07580-f001:**
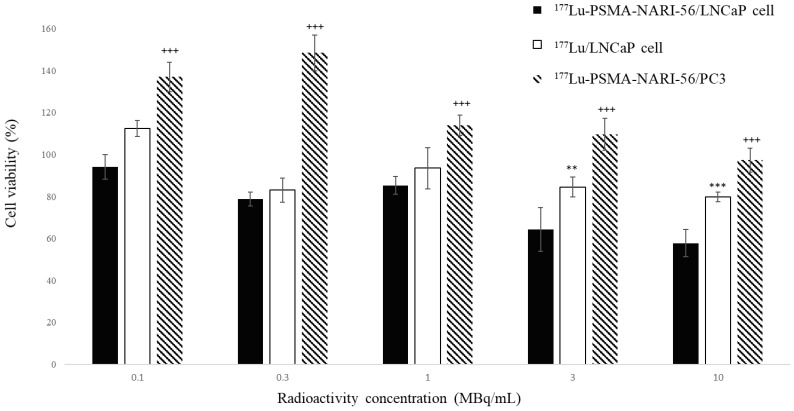
Cell viability of LNCaP and PC-3 cells after exposure at 0.1, 0.3, 1, 3, and 10MBq/mL of ^177^Lu-PSMA-NARI-56 and ^177^Lu. Data are presented as mean ± DEM (n = 8 independent experiments). (** *p* < 0.01 and *** *p* < 0.001 the effects of ^177^Lu-PSMA-NARI-56 compared to ^177^Lu on LNCaP cell; ^+++^ as *p* < 0.001, ^177^Lu-PSMA-NARI-56 on LNCaP cell compared to PC3 cell).

**Figure 2 ijms-26-07580-f002:**
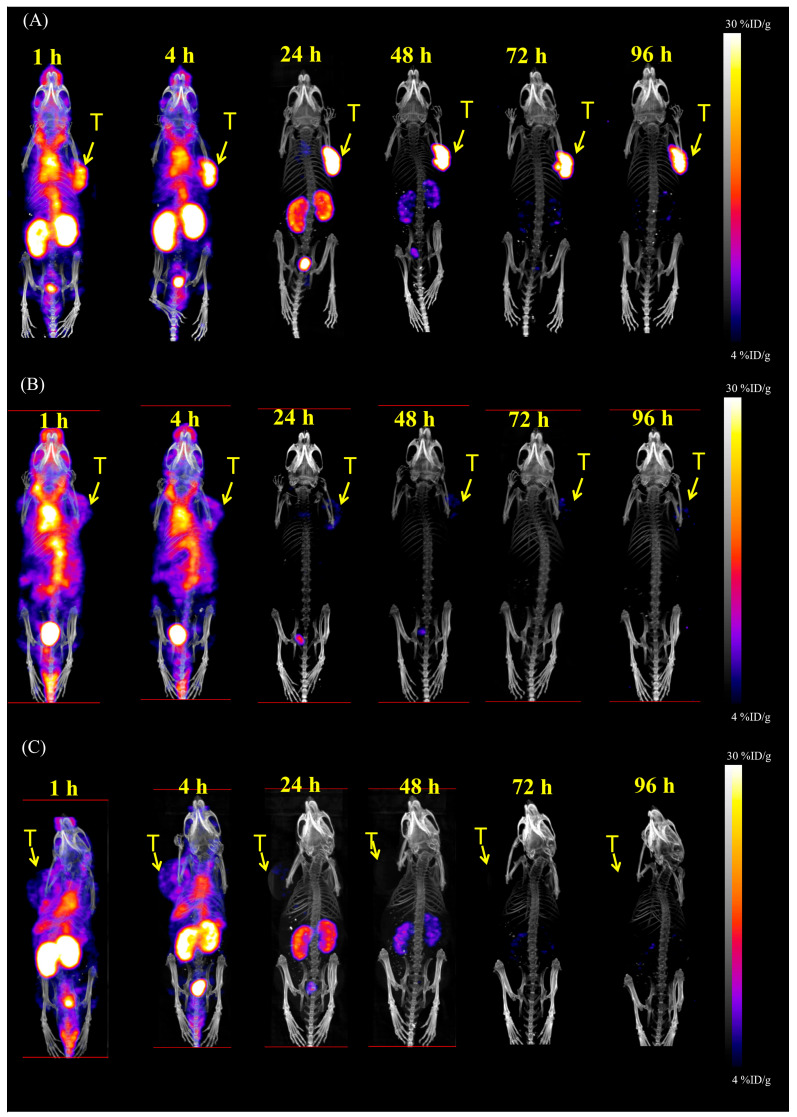
In vivo NanoSPECT/CT static images of ^111^In-PSMA-NARI-56 in LNCaP (**A**,**B**) and PC-3 (**C**) tumor-bearing mice.

**Figure 3 ijms-26-07580-f003:**
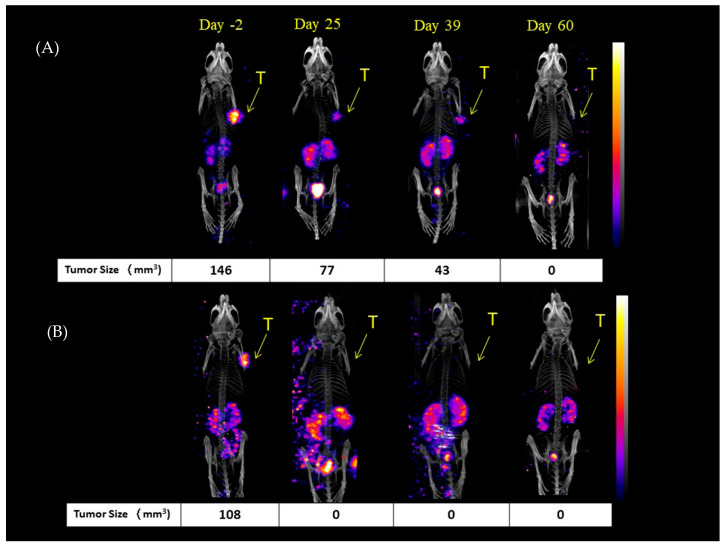
^111^In-PSMA-617/SPECT/CT imaging of LNCaP tumor-bearing mice at days-2, 25, 39, and 60 post-treatment. (**A**) (^177^Lu-PSMA-617) shows progressive tumor reduction from 146 mm^3^ to complete response (0 mm^3^). (**B**) (^177^Lu-PSMA-NARI-56) shows rapid response by day 25 with sustained remission. Fluorescence intensity displayed as pseudocolor scale (yellow = high, purple/blue = low). Arrows indicate tumor locations (T). Values represent tumor volumes in mm^3^.

**Figure 4 ijms-26-07580-f004:**
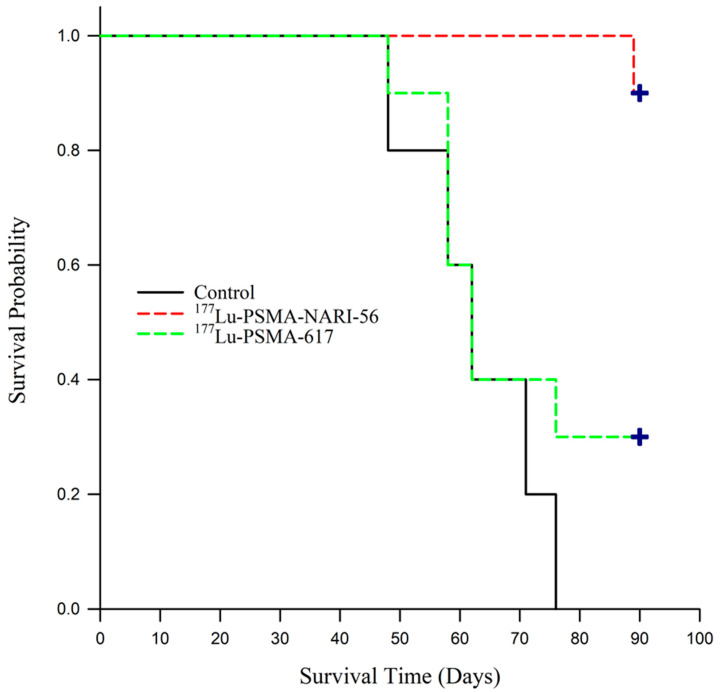
Kaplan–Meier survival analysis of LNCaP tumor-bearing mice. Mice were treated with a single intravenous injection of saline (Control, n = 5), 18.5 MBq of ^177^Lu-PSMA-NARI-56 (n = 10), or 18.5 MBq of ^177^Lu-PSMA-617 (n = 10). Survival differences were assessed using a Log-Rank test, and crosses indicate censored subjects. ^177^Lu-PSMA-NARI-56 significantly improved survival compared to both the Control and ^177^Lu-PSMA-617 groups (*p* < 0.001 and *p* = 0.008, respectively). No significant difference was observed between the ^177^Lu-PSMA-617 and Control groups (*p* = 0.335).

**Table 1 ijms-26-07580-t001:** Biodistribution of 1.85 MBq ^177^Lu-PSMA-NARI-56 in murine LNCaP tumor-bearing male BALB/c nude mice at 1, 4, 24, 48, 72, and 96 h.

Mean ± SD (%ID/g)	Biodistribution of ^177^Lu-PSMA-NARI-56 in LNCaP Tumor-Bearing Mice (n = 4)					
	1 h	4 h	24 h	48 h	72 h	96 h
Blood	27.69 ± 8.21	12.55 ± 1.45	2.13 ± 0.66	0.67 ± 0.30	0.26 ± 0.15	0.13 ± 0.02
Skin	8.85 ± 1.96	3.55 ± 0.54	1.01 ± 0.45	0.35 ± 0.16	0.19 ± 0.08	0.13 ± 0.03
Muscle	3.38 ± 1.15	1.48 ± 0.11	0.28 ± 0.18	0.10 ± 0.04	0.05 ± 0.03	0.07 ± 0.07
Bone	1.98 ± 0.49	0.84 ± 0.12	0.44 ± 0.05	0.28 ± 0.08	0.17 ± 0.09	0.20 ± 0.10
Brain	0.52 ± 0.14	0.30 ± 0.06	0.10 ± 0.05	0.05 ± 0.02	0.03 ± 0.01	0.02 ± 0.00
Bladder	7.74 ± 1.24	7.50 ± 6.70	1.14 ± 0.58	0.26 ± 0.11	0.13 ± 0.06	0.04 ± 0.03
Pancreas	3.48 ± 0.83	1.67 ± 0.41	0.35 ± 0.15	0.12 ± 0.06	0.06 ± 0.02	0.03 ± 0.01
Spleen	9.66 ± 4.22	3.57 ± 1.15	1.14 ± 0.57	0.44 ± 0.17	0.29 ± 0.13	0.16 ± 0.06
Stomach	3.96 ± 1.35	1.75 ± 0.50	0.51 ± 0.30	0.15 ± 0.06	0.06 ± 0.02	0.04 ± 0.01
Small intestine	5.68 ± 0.70	3.21 ± 0.94	0.56 ± 0.27	0.19 ± 0.05	0.08 ± 0.03	0.05 ± 0.02
Large intestine	4.34 ± 1.19	2.27 ± 0.53	0.46 ± 0.20	0.24 ± 0.09	0.10 ± 0.02	0.08 ± 0.02
Bile	6.63 ± 8.40	2.46 ± 2.13	0.53 ± 0.52	0.46 ± 0.29	0.24 ± 0.20	0.06 ± 0.04
Liver	7.15 ± 1.73	3.01 ± 1.04	0.66 ± 0.16	0.32 ± 0.11	0.16 ± 0.03	0.10 ± 0.02
Adrenal	14.72 ± 4.25	8.50 ± 3.38	2.42 ± 1.35	1.35 ± 0.82	0.86 ± 0.30	0.50 ± 0.34
Kidney	107.65 ± 37.19	92.05 ± 19.62	37.27 ± 20.02	8.13 ± 4.46	3.29 ± 2.47	1.55 ± 0.26
Heart	7.43 ± 2.11	2.75 ± 0.41	0.60 ± 0.31	0.18 ± 0.08	0.08 ± 0.05	0.05 ± 0.01
Lung	28.70 ± 7.1	9.90 ± 2.69	2.47 ± 1.15	0.77 ± 0.33	0.35 ± 0.15	0.19 ± 0.04
Tumor	26.52 ± 12.11	28.91 ± 6.05	40.56 ± 10.01	31.32 ± 10.37	18.57 ± 6.02	13.41 ± 2.89

**Table 2 ijms-26-07580-t002:** Tumor volume ratio post-treatment (%) of 1.85 MBq ^177^Lu-PSMA-NARI-56 and ^177^Lu-PSMA-617 in murine LNCaP tumor-bearing BALB/c nude mice at 0–58 days.

Mean ± SD (%)	Tumor Volume Ratio Post-Treatment/Pre-Treatment		
Day of Injection	Control	Lu-177-PSMA-NARI-56	Lu-177-PSMA-617
0	100 ± 0	100 ± 0	100 ± 0
7	152 ± 44	73 ± 23	72 ± 27
10	196 ± 54	46 ± 19	75 ± 34
13	215 ± 66	38 ± 14	71 ± 34
16	248 ± 64	34 ± 11	72 ± 36
20	282 ± 63	36 ± 13	87 ± 66
23	337 ± 82	32 ± 15	93 ± 68
27	358 ± 100	23 ± 12	86 ± 66
30	368 ± 133	22 ± 13	106 ± 102
34	408 ± 150	20 ± 13	145 ± 129
37	437 ± 137	20 ± 15	153 ± 129
41	483 ± 157	17 ± 15	199 ± 185
44	480 ± 175	19 ± 17	217 ± 121
48	594 ± 232	18 ± 14	266 ± 229
51	763 ± 314	17 ± 16	288 ± 281
55	758 ± 313	21 ± 25	299 ± 319
58	906 ± 400	31 ± 39	380 ± 446

## Data Availability

The datasets generated and analyzed during the current study are not publicly available due to an ongoing patent application and potential future technology transfer for clinical development. However, the data are available from the corresponding author upon reasonable request.

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
