# Peer review of "PSMA-Targeted Radiolabeled Peptide for Imaging and Therapy in Prostate Cancer: Preclinical Evaluation of Biodistribution and Therapeutic Efficacy"

_ijms, 2025, doi:10.3390/ijms26157580_

Round 1
Reviewer 1 Report
Comments and Suggestions for Authors
The authors present their research on the evaluation of a PSMA targeting radioligand with an attached albumin binding moiety. Similar agents have previously been investigated but the direct comparison to PSMA-617 in both LNCaP and PC3 is a good inclusion to the body of work for PSMA-albumin binding agents. I would prefer that the authors include the structure of 177Lu-PSMA-NARI-56 so it can be compared to the agents evaluated previously in Mol. Pharmaceutics 2018, 15, 6, 2297–2306 https://doi.org/10.1021/acs.molpharmaceut.8b00152 as linker selection was key in that paper to get optimal uptake. I think there needs to be a thorough edit of English usage throughout due to minor mistakes in language use. The studies were well designed to evaluate the new agent and demonstrate its potential compared to PSMA-617. The study could have been improved if 177Lu SPECT images were also collected for radiotherapy injections to help determine dosimetry to tumor versus kidney. With those comments and in mind and with the suggested edits listed below, I recommend this manuscript for publication after minor revision.
Introduction
Line 34: “the incidence and mortality rates are increasing upward trend” reads awkwardly, perhaps say “the incidence and mortality rates are in an increasing upward trend”
Line 37: you give stats for the world but then say, “in these regions”; you can remove it.
Line 42: “Since 2004, variety of medicines based on the utilization of these medications is contingent upon as the extent of metastatic tumor burden”, please restate more clearly
Line 59: “many isotopes employed for developed with PSMA-targeting agents as”; Again, rephrase to make the meaning clear. E.g. “Many radioisotopes have been employed in the development of PSMA targeting radiotherapeutics such as: 131I (β-emitter, t1/2 8.03 days), 90Y (β-emitter, t1/2 2.66 days), 177Lu (β-emitter, t1/2 6.65 days), and 225Ac (α-emitter, t1/2 9.95 days)
Line 62-63: The approval of the 68Ga agent PSMA-11 is mentioned but its use and reason for approval is not mentioned. The authors should inform the reader that the gallium agent and two newer fluorine-18 agents (pylarify (piflufolastat) and POSLUMA (flotufolastat)) serves to quantify PSMA expression for enrollment of patients in PSMA radiotherapy and to stage disease progression.
Line 71-83: A mention should be made of some of the drawbacks experienced with Pluvicto, chiefly concerns over high Kidney and Salivary gland uptake that can lead to loss of function in those organs, and if this approach can help to ensure more tumor uptake and less uptake in other organs to maximize therapeutic effect and decrease side effects/toxicities.
Results
Figure 2 Caption: Review wording of caption to make sure usage is correct.
Table 1 Caption and description: Provide n value for each measurement using an asterisk for the ones that had less or more than planned (3 vs 4 measurements). Were only male mice used, please state in description.
For therapy doses were imaging studies conducted? Imaging of 177Lu attempted by SPECT is done for patient dosimetry post injection of the therapy and could provide further dosimetry insight on your agent (see articles like those of Violet et. Al Journal of Nuclear Medicine April 2019, 60 (4) 517-523; DOI: https://doi.org/10.2967/jnumed.118.219352
The data comparing tumor volumes from the 177Lu-PSMA-617 vs 177Lu-PSMA-NARI-56 is presented and compelling, is data concerning kidney uptake or function changes available in the data given the kidney toxicity concern with 177Lu-PSMA-617?
Discussion
Line 188: “In previous study, we synthesized a compound by conjugating an albumin-binding molecule and a DOTA chelator onto PSMA-617.” Please provide the reference and please include a figure that shows the structures of177Lu-PSMA-617 and 177Lu-PSMA-NARI-56 so the reader can understand the changes you have made to modify the structure and the linker and chelator utilized.
Line 251-252: That is true but it is worth mentioning the work with JHU-2545 which is a means to selectively deliver 2-PMPA to kidney and salivary glands (e.g. Nedelcovych et. Al JHU-2545 preferentially shields salivary glands and kidneys during PSMA-targeted imaging. Eur J Nucl Med Mol Imaging 52, 1631–1641 (2025). https://doi.org/10.1007/s00259-024-07044-7)
Line 255-276: Information regarding not just the radioactive dose in each study but also the molar activity (GBq/micromole) would be helpful as low molar activity could inhibit binding of the radioligand.
Comments on the Quality of English LanguagePlease address the English usage mistakes throughout the manuscript as it can distract from the quality of the work performed.
Author Response
|
1. Summary |
|
|
Thank you very much for taking the time to review this manuscript. Please find the detailed responses below and the corresponding revisions/corrections highlighted/in track changes in the re-submitted files.
|
|
|
2. Point-by-point response to Comments and Suggestions for Authors |
|
|
Comments 1: Introduction Line 34: “the incidence and mortality rates are increasing upward trend” reads awkwardly, perhaps say “the incidence and mortality rates are in an increasing upward trend” |
|
|
Response 1: Thank you for pointing this out. We agree with this comment. Therefore, we have change and mark the revisions in red. (Line 34)
|
|
|
Line 37: you give stats for the world but then say, “in these regions”; you can remove it. |
|
|
Response 1: Thank you for pointing this out. We deletion those words.
|
|
|
Line 42: “Since 2004, variety of medicines based on the utilization of these medications is contingent upon as the extent of metastatic tumor burden”, please restate more clearly |
|
|
Response 1: Thank you for pointing this out. We agree with this comment. Therefore, we have modified and clearly on this segment and mark the revisions in red. (Line 44-47)
|
|
|
Line 59: “many isotopes employed for developed with PSMA-targeting agents as”; Again, rephrase to make the meaning clear. E.g. “Many radioisotopes have been employed in the development of PSMA targeting radiotherapeutics such as: 131I (β-emitter, t1/2 8.03 days), 90Y (β-emitter, t1/2 2.66 days), 177Lu (β-emitter, t1/2 6.65 days), and 225Ac (α-emitter, t1/2 9.95 days) Response 1: Thank you for pointing this out. We agree with this comment. Therefore, we have change and mark the revisions in red. (Line 63-70)
|
|
|
Line 62-63: The approval of the 68Ga agent PSMA-11 is mentioned but its use and reason for approval is not mentioned. The authors should inform the reader that the gallium agent and two newer fluorine-18 agents (pylarify (piflufolastat) and POSLUMA (flotufolastat)) serves to quantify PSMA expression for enrollment of patients in PSMA radiotherapy and to stage disease progression. |
|
|
Response 1: Thank you for pointing this out. We agree with this comment. Therefore, we have change and mark the revisions in red. (Line 77-81)
|
|
|
Line 71-83: A mention should be made of some of the drawbacks experienced with Pluvicto, chiefly concerns over high Kidney and Salivary gland uptake that can lead to loss of function in those organs, and if this approach can help to ensure more tumor uptake and less uptake in other organs to maximize therapeutic effect and decrease side effects/toxicities. |
|
|
Response 1: Thank you for pointing this out. We agree with this comment. Therefore, we have change and mark the revisions in red. (Line 96-98)
|
|
|
Results Figure 2 Caption: Review wording of caption to make sure usage is correct. |
|
|
Response 1: Thank you for pointing this out. We agree with this comment. Therefore, we have change and mark the revisions in red. (Line 525-526)
|
|
|
Table 1 Caption and description: Provide n value for each measurement using an asterisk for the ones that had less or more than planned (3 vs 4 measurements). Were only male mice used, please state in description. Response 1: Thank you for pointing this out. Therefore, we have change and mark the revisions in red. (Line 515)
For therapy doses were imaging studies conducted? Imaging of 177Lu attempted by SPECT is done for patient dosimetry post injection of the therapy and could provide further dosimetry insight on your agent (see articles like those of Violet et. Al Journal of Nuclear Medicine April 2019, 60 (4) 517-523; DOI: https://doi.org/10.2967/jnumed.118.219352 Response 1: Thank you for pointing this out. In Figure 3 were used therapy doses for SPECT imaging by In-111-PSMA-617. |
|
|
|
|
|
The data comparing tumor volumes from the 177Lu-PSMA-617 vs 177Lu-PSMA-NARI-56 is presented and compelling, is data concerning kidney uptake or function changes available in the data given the kidney toxicity concern with 177Lu-PSMA-617? |
|
|
Response 1: Thank you for pointing this out. We did not conduct experiments on renal function, but the article uses radiation cumulative dose to simulate the consideration of reducing renal radiation damage in the future. We will also try to use 2-(phosphonomethyl)pentanedioic acid (PMPA) and mannitol currently used in clinical practice to reduce the possibility of renal absorption. Therefore, we have mark the related revisions in green. (Line 211-214, 242-255)
|
|
|
Discussion Line 188: “In previous study, we synthesized a compound by conjugating an albumin-binding molecule and a DOTA chelator onto PSMA-617.” Please provide the reference and please include a figure that shows the structures of177Lu-PSMA-617 and 177Lu-PSMA-NARI-56 so the reader can understand the changes you have made to modify the structure and the linker and chelator utilized. Response 1: Thank you for pointing this out. We refer to: Molecular Imaging for Radiolabeling a PSMA-Targeted Long Circulating Peptide as a Theranostic Agent in Mice Bearing a Human Prostate Tumor. Journal of Medical and Biological Engineering, 2021, 41(3):360-368. We have added on introduction in blue (Line 104-106).
|
|
|
Line 251-252: That is true but it is worth mentioning the work with JHU-2545 which is a means to selectively deliver 2-PMPA to kidney and salivary glands (e.g. Nedelcovych et. Al JHU-2545 preferentially shields salivary glands and kidneys during PSMA-targeted imaging. Eur J Nucl Med Mol Imaging 52, 1631–1641 (2025). https://doi.org/10.1007/s00259-024-07044-7) Response 1: Thanks to the committee's recommendation, Dr. Barbara Slusher, Director of JHDD, will work with JHU faculty Dr. Rangaramanujam, Dr. Kannan, and Dr. Parikh to develop and commercialize JHU-2545 as a kidney and salivary gland shielding agent for patients receiving PSMA radiotherapy. Pre-treatment with JHU-2545 in the 2025 article reduced the uptake of 68Ga-PSMA-11 and 18F-DCFPyL by the kidneys and salivary glands by up to 85% without affecting tumor uptake. If this strategy can also be reproduced in the clinic, it is expected to reduce the risk of non-tumor toxicity in PSMA radiotherapy, while allowing patients to receive higher cumulative radiation doses, improving treatment efficacy and safety. I think this is also our goal for future learning. This reference have add on introduction in blue (Line 83-92). |
|
|
|
|
|
Line 255-276: Information regarding not just the radioactive dose in each study but also the molar activity (GBq/micromole) would be helpful as low molar activity could inhibit binding of the radioligand. |
|
|
Response 1: Thank you for pointing this out. We had list the radioactive (GBq/μmole) at Line315-317 mark the revisions in red.
|
|
|
Comments on the Quality of English Language Please address the English usage mistakes throughout the manuscript as it can distract from the quality of the work performed. Response 1: We have asked Editage company (https://app.editage.com.tw/?type=individual) to revise this article. If there are still a lot of English errors, we will ask the company to revise it again. |
|

Reviewer 2 Report
Comments and Suggestions for Authors
The article titled “PSMA-targeted radiolabeled peptide for imaging and therapy in prostate cancer: Preclinical evaluation of biodistribution and therapeutic efficacy” presents a preclinical evaluation of 177Lu-PSMA-NARI-56, a new albumin-binding PSMA-targeted radioligand, demonstrating significantly improved therapeutic efficacy over 177Lu-PSMA-617 in LNCaP tumor-bearing mice. The study design addresses cell viability, biodistribution, imaging, and therapeutic outcomes, with compelling data showing 98% tumor growth inhibition and 90% survival at 90 days post-treatment compared to 58% inhibition and 30% survival with the control agent. The rationale for albumin-binding modification to enhance tumor uptake is well-supported by biodistribution results revealing 40.56±10.01%ID/g tumor uptake at 24 hours and prolonged retention. However, the high kidney uptake (107.65±37.19%ID/g at 1 hour) remains a concern despite the authors' argument that murine kidney PSMA expression differs from humans. While the survival data is impressive, the single-dose design without dose-escalation studies limits clinical translatability. The statistical analysis relies solely on Student's t-test without addressing multiple comparisons, and the absence of body weight/hematological toxicity data in therapy studies weakens safety claims. The manuscript also lacks direct comparison of biodistribution between 177Lu-PSMA-NARI-56 and 177Lu-PSMA-617 under identical conditions, making quantitative improvements difficult to assess. Although SPECT/CT imaging confirms tumor targeting, quantitative ROI analysis would strengthen the imaging conclusions. The discussion appropriately contextualizes the results within existing literature but overstates clinical readiness without larger animal models or toxicity profiles. Technical details like specific activity calculations and purification methods require clarification, and typographical errors (e.g., "deescalated" instead of "castration-resistant") necessitate thorough proofreading. Despite these limitations, the therapeutic superiority demonstrated warrants further investigation. The reviewer has the following comments that authors need to address:
- The selection of the 18.5 MBq dosage in the therapy studies would benefit from additional justification beyond the current dose escalation data. Providing a clear rationale such as referencing prior preclinical efficacy, safety profiles, or established therapeutic windows would strengthen the manuscript by helping readers understand the basis for this dosage choice and its relevance to achieving optimal therapeutic outcomes.
- To strengthen the safety claims of the therapy studies, it is recommended to include detailed toxicity monitoring data. Specifically, reporting body weight trends, key hematological parameters, and renal function markers throughout the treatment period will provide comprehensive insight into the systemic effects of the therapy and better support the conclusions regarding its safety profile. Including these measures will enhance the robustness and credibility of the study.
- For a more comprehensive overview of PSMA-targeted strategies in prostate cancer, the reviewer recommends citing relevant recent studies that investigate selective PSMA targeting approaches to enhance delivery and therapeutic outcomes. These works contribute important insights into optimizing targeting specificity and therapeutic potential in prostate cancer models. Including such references in the introduction will enrich the background and underscore the significance of PSMA-directed imaging and therapy approaches discussed in this manuscript.
https://www.cell.com/molecular-therapy-family/nucleic-acids/fulltext/S2162-2531(24)00080-5
https://www.mdpi.com/1422-0067/25/4/2123
- It is important that the authors provide a clear and specific reference number to support the statement, “In our previous study, we effectively synthesized a PSMA-targeting compound known as PSMA-NARI-56”.
- Including a direct side-by-side comparison of biodistribution data for 177Lu-PSMA-NARI-56 and 177Lu-PSMA-617 at identical timepoints would significantly strengthen the manuscript by providing clearer insights into their relative in vivo behaviors and therapeutic potential.
- Incorporating semi-quantitative tumor-to-background ratios into the SPECT/CT analysis would enhance the imaging data by providing more precise and comparative measures of tracer uptake, thereby improving the evaluation of targeting specificity and diagnostic value.
- Clarification on the purification methods employed after radiolabeling and detailed explanation of how specific activity was calculated would improve the technical rigor and reproducibility of the study.
- Adding preliminary data on nephroprotective agents such as lysine or arginine would enhance the discussion on toxicity mitigation. Additionally, more prominently acknowledging the limitations arising from differences in PSMA expression between murine models and humans would provide a balanced perspective and strengthen the manuscript.
- Typographical errors such as "deescalated" in the Introduction and inconsistencies in timepoints (for example, 48 hours versus 96 hours in Table 1) should be addressed to enhance the manuscript’s clarity.
Author Response
For research article
|
Response to Reviewer II Comments
|
||
|
1. Summary |
|
|
|
Thank you very much for taking the time to review this manuscript. Please find the detailed responses below and the corresponding revisions/corrections highlighted/in track changes in the re-submitted files.
|
||
|
2. Point-by-point response to Comments and Suggestions for Authors |
||
|
Comments 2: 1. The selection of the 18.5 MBq dosage in the therapy studies would benefit from additional justification beyond the current dose escalation data. Providing a clear rationale such as referencing prior preclinical efficacy, safety profiles, or established therapeutic windows would strengthen the manuscript by helping readers understand the basis for this dosage choice and its relevance to achieving optimal therapeutic outcomes. Response 2: Thank you for pointing this out. Therefore, we have added the revisions in blue. (Line 278-284)
|
||
|
2. To strengthen the safety claims of the therapy studies, it is recommended to include detailed toxicity monitoring data. Specifically, reporting body weight trends, key hematological parameters, and renal function markers throughout the treatment period will provide comprehensive insight into the systemic effects of the therapy and better support the conclusions regarding its safety profile. Including these measures will enhance the robustness and credibility of the study. Response 2: Thanks to the committee members for the suggestions. Currently, Lu-177-PSMA-NARI-56 has been listed as one of the key development priorities. According to the : Safety Standards for Non-Clinical Trials of Drugs. The toxicology at least need two mammals, one rodent and the other non-rodent, should be used for testing. The test is currently underway and the results will be published in the form of academic articles in the future to provide complete toxicology monitoring results.
|
||
|
3. For a more comprehensive overview of PSMA-targeted strategies in prostate cancer, the reviewer recommends citing relevant recent studies that investigate selective PSMA targeting approaches to enhance delivery and therapeutic outcomes. These works contribute important insights into optimizing targeting specificity and therapeutic potential in prostate cancer models. Including such references in the introduction will enrich the background and underscore the significance of PSMA-directed imaging and therapy approaches discussed in this manuscript. https://www.cell.com/molecular-therapy-family/nucleic-acids/fulltext/S2162-2531(24)00080-5 Response 2: Thank you for pointing this out. Therefore, we have added some reference in blue. (Line 83-92, 104-106)
|
||
|
4. It is important that the authors provide a clear and specific reference number to support the statement, “In our previous study, we effectively synthesized a PSMA-targeting compound known as PSMA-NARI-56”. Response 2: Thank you for pointing this out. Therefore, we have added the reference in blue. (Line 104-106) and added supplement 1 (Line 187) file as the structure of two compounds.
|
||
|
5. Including a direct side-by-side comparison of biodistribution data for 177Lu-PSMA-NARI-56 and 177Lu-PSMA-617 at identical timepoints would significantly strengthen the manuscript by providing clearer insights into their relative in vivo behaviors and therapeutic potential. Response 2: Thank you for pointing this out. Therefore, we have added a sentence in the conclusions with blue. (Line 409-414).
|
||
|
6. Incorporating semi-quantitative tumor-to-background ratios into the SPECT/CT analysis would enhance the imaging data by providing more precise and comparative measures of tracer uptake, thereby improving the evaluation of targeting specificity and diagnostic value. Response 2: Thank you for pointing this out. Therefore, we had modified Discussion section with SPECT/CT analysis ratios about PSMA-NARI-56 specificity and diagnostic value. And the modified sentence with brown. (Line 201-227).
|
||
|
7. Clarification on the purification methods employed after radiolabeling and detailed explanation of how specific activity was calculated would improve the technical rigor and reproducibility of the study. Response 2: Response 2: Thank you for pointing this out. Therefore, we have added the reference in blue. (Line 104-106) and added supplement 2 (Line 189) file as the radiolabeling and radiochemical identification 177Lu-PSMA-NARI-56.
|
||
|
8. Adding preliminary data on nephroprotective agents such as lysine or arginine would enhance the discussion on toxicity mitigation. Additionally, more prominently acknowledging the limitations arising from differences in PSMA expression murine models and humans would provide a balanced perspective and strengthen the manuscript. Response 2: Thanks to the committee's recommendation. A new develop and commercialize JHU-2545 as a kidney and salivary gland shielding agent for patients receiving PSMA radiotherapy. Pre-treatment with JHU-2545 in the 2025 article reduced the uptake of 68Ga-PSMA-11 and 18F-DCFPyL by the kidneys and salivary glands by up to 85% without affecting tumor uptake. If this strategy can also be reproduced in the clinic, it is expected to reduce the risk of non-tumor toxicity in PSMA radiotherapy, while allowing patients to receive higher cumulative radiation doses, improving treatment efficacy and safety. I think this is also our goal for future learning. This reference has added on Discussion (Line 242-255) as green words.
9. Typographical errors such as "deescalated" in the Introduction and inconsistencies in timepoints (for example, 48 hours versus 96 hours in Table 1) should be addressed to enhance the manuscript’s clarity. Response 2: I am not quite sure about this part. If the problem has not been solved yet, please give me some advice.
|
||
|
4. Response to Comments on the Quality of English Language |
||
|
Response 2: We have asked Editage company (https://app.editage.com.tw/?type=individual) to revise this article. If there are still a lot of English errors, we will ask the company to revise it again. |
||
